# Antidiarrheal, analgesic,antidepressant, antimicrobial and hypoglycemic activities of methanolic extract from *Sonneratia apetala* fruit, with identification of bioactive compounds in n-hexane, chloroform, and ethyl acetate fractions

Fatema Akhter[1], [2], Md. Ripaj Uddin[3], *, Muhammad Abdullah Al Mansur[3], Mohammad Saydur Rahman[2], Md. Ahedul Akbor[3], Nahida Akter[3], Abubakr M. idris[4], Md. Golam Mostafa[5], AHM Shofiul Islam Molla Jamal[3], Sarker Kamruzzaman[3]

**1** Department of Pharmacy, Bangladesh University, Dhaka, Bangladesh, **2** Department of Pharmacy, Jagannath University, Dhaka , Bangladesh, **3** Institute of National Analytical Research and Service (INARS), BCSIR, Dhanmondi, Dhaka , Bangladesh, **4** Department of Chemistry, King Khalid University, College of Science, Abha, Saudi Arabia, **5** Institute of Mining, Mineralogy and Metallurgy, BCSIR, Joypurhat, Bangladesh

* md.ripajuddin@gmail.com (MRU)

## Abstract

The rapidly growing mangrove fruit *Sonneratia apetala*, native to the deltaic region of Bangladesh, holds promise in traditional medicine due to its bioactivity and antimicrobial properties. Sample collections from Nijum Dwip, Hatiya, Noakhali in Bangladesh were divided into pericarps and seeds, subsequently fractionated with methanol, n-hexane, ethyl acetate, and chloroform. Bioactivity assays involved Swiss albino mice, acquired from ICDDR, B, in compliance with FELASA standards. Standard agents such as diclofenac sodium, loperamide, diazepam, and glibenclamide were used to evaluate antidiarrheal, antidepressant, hypoglycaemic, and analgesic effects, while ciprofloxacin served as a reference for antibacterial and antifungal testing. Methanolic extracts (ME) of the seed and pericarp exhibited notable peripheral and central analgesic effects at 200 and 400 mg/kg dosages. The ME of seeds demonstrated the strongest antidiarrheal efficacy at 400 mg/kg after 1 hour, and the pericarp at 200 mg/kg after 2 hours. The ME also showed significant antidiabetic potential in both seed (99%) and pericarp extracts. GC-MS analysis disclose seven bioactive compounds in the n-hexane, ethyl acetate, and chloroform fractions, including N-Ethyl-2-methylbenzenesulfonamide, 3,6-Pyridazinedione, 1,2-dihydro-1-(4-nitrophenyl), N-Acetyl-alpha-aminooxybutyric acid (methyl), 2H-Phenanthro[2,1-b] azepin-2-one (1,3,4,5,5a), and Undecane. These compounds have established anticancer and antimicrobial properties. Both pericarp and seed extracts displayed strong antifungal activity against *Saccharomyces cerevisiae*, *Candida albicans*, and *Aspergillus niger*, while moderate antibacterial effects were noted against gram-negative strains like *Pseudomonas aeruginosa* and *Salmonella*

**Data availability statement:** All relevant data are within the paper and its Supporting Information files.

**Funding:** The author(s) received no specific funding for this work.

**Competing interests:** The authors declare no competing interests.

*Typhi* as well as gram-positive bacteria such as *Staphylococcus aureus*. These findings underscore *S. apetala*'s potential as a valuable bioactive source for traditional medicinal applications.

---

## 1. Introduction

*Sonneratia apetala* (Buch. Ham.) is a fast-growing square-branched mangrove commonly found along the coasts of Australia, Bangladesh, Malaysia, and India, among other countries [1]. They are key parts of natural ecosystems and develop in recently accreted soil with moderate to high salinity [2]. *S. apetala* fruit have a resilient leathery calyx, and its diameter ranges from 1.5 to 2 cm, and is plentiful in vitamins, especially Vitamin C, minerals, and a variety of other nutrients, making its edible portions (the fleshy part of the fruit) around 20% of the fruit. Each fruit consists of pericarp and seeds (Fig 1). Those seeds are concentrated into a single large seed, and the pericarp is very small. The seeds are yellowish in color and are packed closely together in 6 to 8 locules primarily in a U or V shape [3]. The ME of *S. apetala* has been reported to contain polyphenols, anthocyanins, flavonoids, and Vitamin C. Functional components such as polyphenols, flavonoids, and anthocyanins with the extract of spiked fruit offering a high content of these compounds are responsible for its potential antibacterial properties [4]. While communities along the coast use *Sonneratia apetala* fruit juice as a tonic to cure various ailments including diarrhoea [5]. *S. apetala* has shown antimicrobial, antioxidant, and cytotoxic activities but, to the best of our knowledge, no extract of its green fruit has been used for catalytic synthesis of arylidene-malononitrile. This fruit's aqueous extract being 100% eco-friendly and bio-degradable makes this part promising candidate for green chemistry applications as catalyst.

Diarrhea is still a critical public health concern, especially in developing regions with insufficient availability of clean water and correct waste disposal. Plant-based treatments have been shown to have great potential against diarrhoea, and *S. apetala* is a plant that shows such potential. *S. apetala* fruit ethanolic extract has previously been reported to anhypo-diarrhoeal in an animal model and has been demonstrated to mediate its effect mainly through modulation of gut motility and decreased intestinal secretion [6]. This antidiarrheal action is thought to be related to the astringent action of tannins and flavonoids, which reduce fluid loss through the intestine and improve intestinal health [7]. It is important to note that depression is a common and impairing mental health condition that affects millions of people worldwide. Despite pharmaceutical interventions being available, their side effects have led to a burgeoning interest in alternative therapies. Antidepressant-like activity of *S. apetala* fruit ethanolic extract in animal models may improve the levels of dopaminergic and serotonergic neurotransmitters [1]. It is postulated that these bioactive compounds present in *S. apetala* are responsible for working on the central nervous system, hence impacting the regulating of mood. This indicates that the fruit extract may be a potential candidate for drug development as a natural antidepressant agent with little side effects than those of

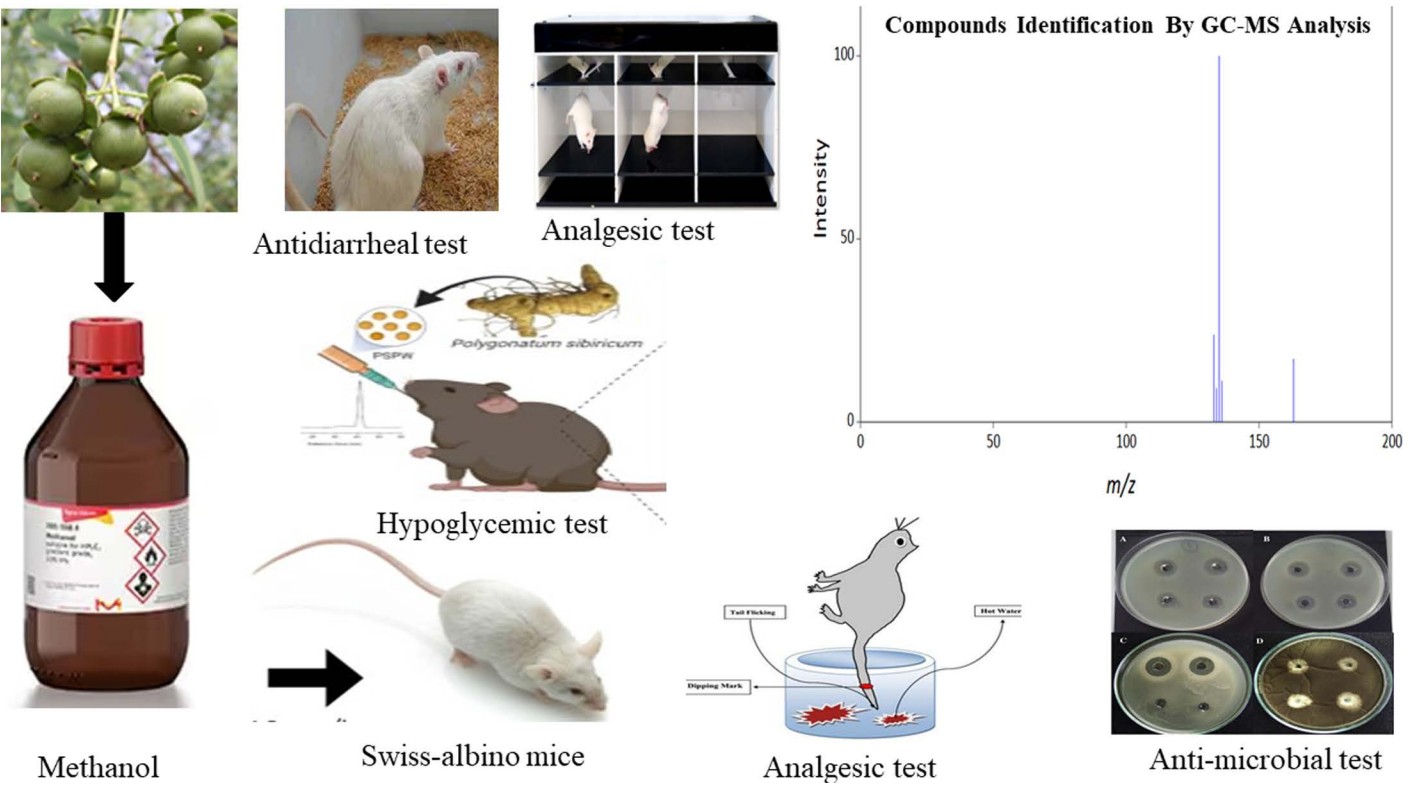

**Fig 1. Experimental Design for different crude fraction of *S. apeta* fruit extracts.**

the conventional treatments. Background: Diabetes mellitus, particularly type 2 diabetes, has emerged as a significant and growing public health issue worldwide. It is important to control blood glucose level as the management of such pleiotropic diseases. *S. apetala* fruit that demonstrated hypoglycemic activity in rats likely through enhanced insulin secretion and diminished oxidative status [8]. Contiguous experimental studies indicate that treatment of diabetic animal models with ME extract causes significantly reduced blood glucose levels showing its potential utility as a natural therapeutic for diabetes management. The hypoglycemic effects are attributed to bioactive compounds found in the fruit including saponins and flavonoids [1]. Regardless, pain management is an important target of research because existing analgesic medicines, including opioids, can be addictive and have adverse side effects. The ethanolic reads of *S. apetala* fruit exhibit analgesic activity in animal models, probably due to the inhibition of prostaglandin synthesis, a significant mechanism in inflammation and pain perception [9]. The smear technique also reinforced the potential for the extract to act as a natural analgesic agent that could be used in place of synthetic painkillers that carry more side effects. Over the past several years, there has been an increasing demand for alternatives for the battle against bacterial and fungal infections because of the rise in antimicrobial resistance. *S. apetala* fruit showed significant antimicrobial activity against various pathogenic bacteria and fungi [1]. The antimicrobial activity is probably due to the bioactive components such as alkaloids, tannins, and flavonoids which have been shown to damage the cell wall of the micro-organism and prevent their proliferation [10]. This showcases *S. apetala* as an attractive reservoir of novel antimicrobial compounds that could help tackle the global threat of antimicrobial resistance.

Medical anthropological studies have demonstrated the significant value of plant-based medicine in treating illnesses and improving lives, much like conventional medicines. Given that plants offer a rich source of bioactive natural compounds, surpassing those found in animals, many individuals have long turned to alternative medicine as

a soothing alternative to traditional treatments. Bangladesh, with its diverse array of medicinal plants spanning different families, presents a valuable resource. Consequently, it becomes imperative to conduct rigorous scientific investigations aimed at discovering and developing new, safer medicinal options for addressing various health conditions. As per the available literature, there have been limited reports regarding the analgesic, hypoglycemic, antidiarrheal, antidepressant, and antimicrobial activities of different fractions derived from this plant. Hence, this research holds immense potential and warrants considerable attention. The main goals of this research are to evaluate the biological effects, such as analgesic, hypoglycemic, antidiarrheal, antidepressant, and antimicrobial properties, through live testing of the soluble part of the methane extract (MESF) derived from both the outer shell and seeds of S. apetala. Additionally, the study aims to identify the active components in the n-hexane, ethyl acetate, and chloroform portions using the GC-MS method.

## 2. Materials and methods

### 2.1 Ethics approval statement

The Ethics Committee of State University of Bangladesh approved the study protocol on 2022-01-12 (SUB/A-ERC/005).

### 2.2 Sample collection, identification, and authentication procedures

In August 2021, *S. apetala* fruits were collected from deltaic Island Hatia, Noakhali, Bangladesh [11]. The fruits part such as pericarps and seeds were separated, dried, ground into fine powder, and kept in airtight containers. A voucher sample (DACB No. 66955) was verified and stored by an expert at the Bangladesh National Herbarium in Dhaka.

### 2.3 Experimental design, extraction and processing

The powdered pericarp (250 g) of *S. apetala* was immersed in 2.5 L of methanol for 72 hr. at room temperature, with intermittent shaking and stirring, adhering to a standardized protocol [1].

The crude extract was filtered using cotton and Whatman (No. 1) paper, then concentrated by evaporating the solvent at low pressure and under 40°C with a rotary evaporator. An identical procedure was employed for the dried, powdered seeds (250 g), which were also soaked in methanol for 72 hours. The resulting extracts, including methanol, n-hexane, ethyl acetate, and chloroform fractions, were separated via filtration and solvent evaporation under reduced pressure. Approximately 10 g of each extract was stored under refrigeration for subsequent analysis. The methanol extracts (ME) derived from both the seed and pericarp were assessed for their analgesic, antidiarrheal, antidepressant, and antibacterial properties. Bioactive compounds within the n-hexane, ethyl acetate, and chloroform fractions were identified using GC-MS. The experimental workflow for the various fractions is illustrated in Fig 1.

### 2.4 Experimental animals

Swiss-albino mice, aged between four and five weeks, weighing approximately 23 to 25 grams, and comprising both sexes [12], were utilized in the experiment (see Fig 2). These mice were procured from the International Centre for Diarrhoeal Diseases and Research (ICDDR, B) in Bangladesh. To ensure proper acclimation, each animal underwent a week of adjustment before the commencement of the studies. During the experiments, the mice received a nutritionally complete diet from icddr,b and had unlimited access to tap water. The specimens were maintained under optimal laboratory conditions, with 55–65% humidity, a controlled 25.0°C temperature, and a 12-hour light/dark cycle to ensure consistency. Prior to each experiment, the mice underwent a 12-hour fasting period to ensure standardised conditions. All procedures adhered to the Federation of European Laboratory Animal Science Associations (FELASA) guidelines to reduce stress and discomfort.

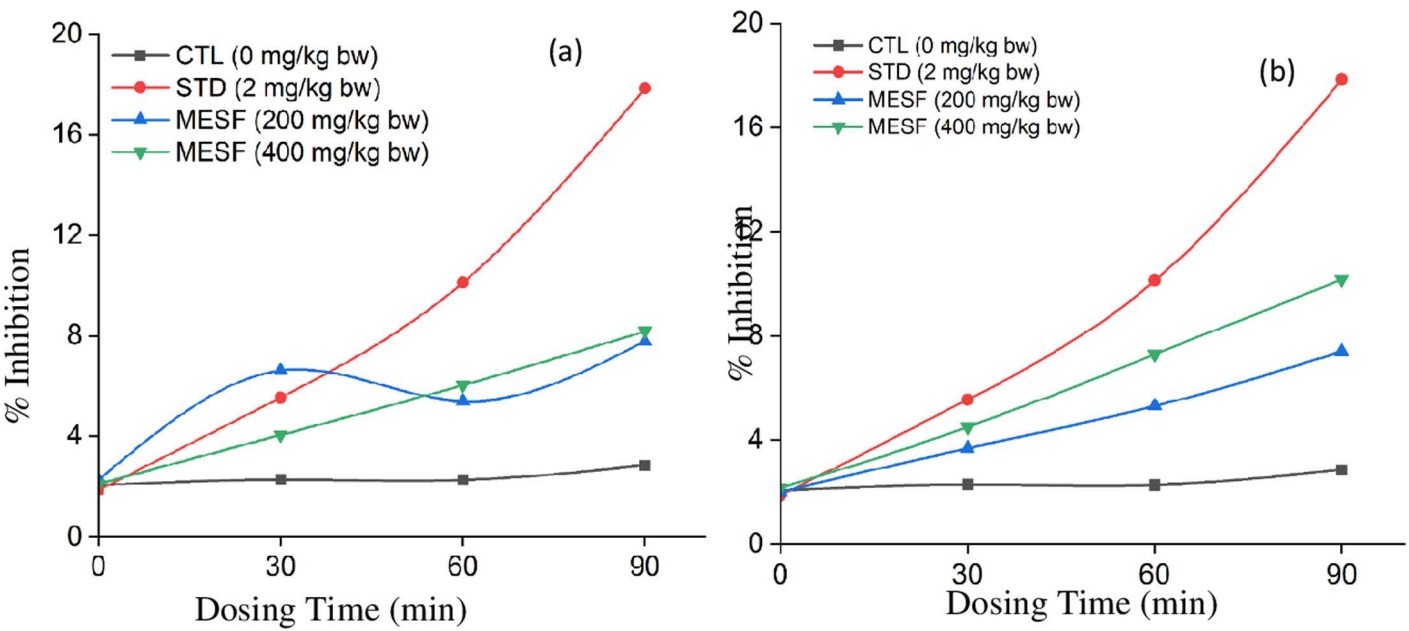

**Fig 2. Assessment of central analgesic activity in crude extract from *S* (a) pericarp and (b) seed of *S. apetala*.**

## 2.5 Experimental model drugs

The medications used in this study included loperamide, glibenclamide, diazepam, and diclofenac sodium, which were obtained from local pharmaceutical companies: Opsonin Pharma Ltd., Square Pharmaceuticals Ltd., and Gonoshasthaya Pharmaceuticals Ltd., respectively [13].

## 2.6 Central analgesic activity properties of ME

The central analgesic effect was assessed using the tail immersion method [9]. Mice were separated into four equal groups: the standard group received diclofenac sodium (5mg/kg bw), the control group was given 1% Tween 80 (10ml/kg bw), and the test groups were administered crude extract doses of 200mg/kg and 400mg/kg. During the experiment, the distal 1–2cm of each mouse's tail was immersed in water heated to 55°C, and the time taken for the tail to be withdrawn was recorded with a stopwatch. Reaction times were measured at 0, 30, 60, and 90 minutes post-administration of the test substances, with a maximum latency of 15 seconds. Baseline reaction times were taken before the treatments were given to establish a reference for comparison.

## 2.7 Peripheral analgesic activity test of ME

The writhing test, induced by acetic acid, is used to assess the peripheral analgesic properties of the subjects. Writhing is characterized by body contractions and squirming, which occur as long as the animals experience discomfort or unpleasant sensations [14]. Mice were divided into eight groups, with each group containing four animals. The standard group was given diclofenac sodium (5mg/kg bw), the experimental groups received crude extract doses of 200 and 400mg/kg bw and the control group received 1% Tween 80 in saline (10ml/kg bw). All mice were intraperitoneally injected with 1% acetic acid, maintaining a 30-minute interval between injections. Group I functioned as the primary experimental group, while Group III acted as the control. The positive control group was treated with diclofenac sodium (5mg/kg); control group was received normal saline containing 1% Tween 80. Test samples dissolved in normal saline containing 400mg/kg Tween 80 were given to the experimental group. The acetic acid injection and oral administration of the test substances

were separated by an interval of 30 min to allow absorption. The number of writhing movements per mouse was counted for ten minutes after injection of acetic acid, and the onset and length of sleep were recorded for both the control and treatment groups.

## 2.8 Antidiarrheal activity test of ME

Antidiarrheal activity of MEs from *S. apetala* pericarp and seeds were studied against castor oil induced diarrhea in Swiss albino mice [1]. Sixteen mice (n = 4 per group) were randomly divided into four groups. The experimental groups were treated with same 200 and 400 mg/kg bw of extract, while as the control group received normal saline containing 1% Tween 80 (10 ml/kg bw). The control was treated with loperamide (50 mg/kg bw). The mice were individually housed in paper-lined cages for collection of fecal samples. To induce diarrhoea, each mouse was given 1 mL of castor oil orally with extracts and loperamide given one hour earlier. Under the five-hour faecal output using nociceptive stimuli observation and measurement.

## 2.9 Antidepressant activity test of ME

For the thiopental sodium-induced sleep time test, a modified version of the method [15] was utilised. The study comprised four groups of four mice each. The positive control group was treated with benzodiazepines as the standard, while the test groups received extract doses of 200 and 400 mg/kg bw. The control group was given saline water with 1% Tween 80. Sleep onset and duration were observed for both the control and test groups.

## 2.10 Hypoglycemic activity test of ME

A glucose tolerance test (GTT) was performed to evaluate hypoglycemic activity. Four- to five-week-old Swiss albino mice (23–25 g) of both sexes were obtained from the ICDDR, B. Mice were randomly assigned to four groups of sixteen, with their weights measured before treatment to adjust the dosages of test and control substances accordingly. The MEsof 60 mg and 30 mg were precisely weighed and then diluted in 1 ml of distilled water. A single drop of the suspending agent Tween 80 was added to the mixture. The crude extracts were dissolved in distilled water using a vortex mixer. A precise volume of 0.2 ml of the test solution was administered to each mouse in the experimental groups using a feeding needle. For the standard treatment, a 10 mg glibenclamide tablet was dissolved in 3.0 ml of normal saline (0.9% NaCl). After a 60-minute interval, all groups received a 10% glucose solution. Blood samples were drawn from the tail vein at 120 and 180 minutes post-glucose administration, and blood glucose levels were assessed using a glucometer.

## 2.11 Antimicrobial screening of ME

Antimicrobial activity was evaluated using the disc diffusion method [16,17]. Sterile discs were loaded with 400 µg of the diluted samples, while ciprofloxacin (30 µg/disc) was used as the standard antibiotic for both antibacterial and antifungal tests. The minimum inhibitory concentration (MIC) of ciprofloxacin ranges from 0.015 to 2 µg/mL.

### 2.11.1 Test organisms, culture medium and composition of Nutrient agar medium.
The bacterial strains employed in this research were obtained as pure cultures from the Institute of Nutrition and Food Science (INFS), University of Dhaka, encompassing both Gram-positive and Gram-negative bacteria, along with fungi. Gram-negative strains comprised *Vibrio mimicus*, *Escherichia coli*, *Pseudomonas aeruginosa*, *Shigella dysenteriae*, *Shigella boydii*, *Salmonella Paratyphi*, *S. Typhi*, and *V. parahemolyticus*; Gram-positive strains included *Bacillus subtilis*, *Bacillus megaterium*, *Staphylococcus aureus* and *Sarcina lutea*; and fungal strains such as *Candida albicans*, *Saccharomyces cerevisiae*, and *Aspergillus niger* were used for performance evaluation. Nutrient agar medium was prepared for conducting antibacterial and antifungal sensitivity tests, as well as for cultivating fresh cultures. The medium consisted of the following components per litre: sodium chloride (5.0 g), peptone (5.0 g), beef extract (1.5 g), yeast extract (1.5 g), and agar (15.0 g).

**2.11.2 Preparation of the culture medium, sub culturing, and the setup of test plates and sample discs.** The culture medium was prepared by dissolving precise quantities of each constituent in distilled water within a conical flask, achieving the target volume. The solution was heated in a water bath until clarity was attained, followed by pH adjustment to 7.2–7.6 (at 25°C) using NaOH or HCl. Aliquots of 10 mL and 5 mL were dispensed into screw-cap test tubes for plate and slant preparation, respectively. The tubes were securely sealed and sterilized via autoclaving at 121°C and 15 psi for 20 minutes. The slants were then used for cultivating fresh bacterial cultures for sensitivity testing. Under aseptic conditions in a laminar airflow cabinet, pure cultures of the test organisms were transferred to agar slants using a sterilized transfer loop to establish fresh cultures. The cultures were incubated at 37°C for 24 hours to facilitate microbial growth, followed by utilizing fresh cultures for sensitivity analysis. Sterilized loops were employed to inoculate approximately 10 mL of agar medium in test tubes under aseptic conditions. The tubes were gently rotated to ensure uniform suspension of the organisms, which was then promptly poured into sterilised Petri dishes. Each dish was rotated clockwise and counterclockwise multiple times to achieve an even distribution of organisms within the medium.

Three types of discs were employed in the antibacterial screening: standard antibiotic discs as positive controls to verify activity against test organisms and compare with a known antibacterial agent, ciprofloxacin discs as the reference standard, and blank discs as negative controls to confirm that no residual solvent or the filter paper exhibited antimicrobial activity. Under aseptic conditions, test samples were prepared by dissolving a precise quantity of each compound in a defined volume of solvent to reach the required concentration. The discs were then soaked in these sample solutions and air-dried before use.

**2.11.3 Preparation of sample discs with test compounds involved diffusion, incubation, and sterilization procedures.** The antimicrobial and antifungal properties of crude extracts and purified compounds were evaluated against a range of Gram-positive, Gram-negative bacteria, and fungal species. For crude extract analysis, 400 µg of each sample was uniformly applied to individual discs, with the same concentration maintained for pure compound testing. The sample discs, along with standard antibiotic and control discs, were precisely positioned on agar plates pre-seeded with target microbial strains. The plates were refrigerated at 4°C for 24 hours to facilitate optimal diffusion of the compounds into the agar medium. Subsequently, the plates were inverted and incubated at 37°C for an additional 24 hours to assess inhibitory activity. All antimicrobial screenings were conducted in a Laminar Flow Hood under aseptic conditions to prevent contamination. The UV light in the hood was activated for one hour before starting. Petri dishes and glassware were sterilized in an autoclave at 121°C and 15 psi for 20 minutes. Additionally, micropipette tips, forceps, cotton, and blank discs were sterilized before use.

## 2.12 GC–MS analysis for unknown compounds identification

GC-MS analysis was conducted using a GC-MS-QP 2010 Ultra system fitted with an AB Innowax capillary column (30 m × 0.25 mm internal diameter, 0.25 µm film thickness). A 0.5 µL sample volume was injected, optimized for sensitivity, column capacity, stability, instrument efficiency and analyte concentration. The oven started at 120°C, held for 1 minute, then increased to 270°C over 25 minutes. He gas was the carrier gas at 1.15 mL/min with a 200 split ratio. Injector and mass transfer line temperatures were 200°C and 250°C, respectively. Electron impact ionization at 70 eV captured mass spectra for 40.75 minutes, scanning 50–650 m/z. Compounds were confirmed by matching retention indices with Wiley, NIST Libraries, and literature [18].

## 2.13 Statistic evaluation

The result were presented as mean ± SEM values. Statistical assessment was performed using Student's t-test. The findings were evaluated in comparison to the control group and were considered significant at P values <0.05.

# 3. Results and discussion

## 3.1 Analgesic activity of the ME of *S. apetala* fruit

The bioactive potential of ten edible mangrove fruits *Aegiceras corniculatum, Avicennia officinalis, Bruguiera gymnorrhiza, Ceriops decandra, Heritiera fomes, Nypa fruticans, Phoenix paludosa, Sarcolobus globosus, Sonneratia caseolaris, and Xylocarpus mekongensis* from the Sundarbans, the largest mangrove forest in the world, reveals significant anti-bacterial and anti-diarrheal properties. These findings highlight the potential of these fruits as valuable sources for pharmaceutical and nutraceutical development [19]. Fig 2 and Table S1 presents the percentage increase in elongation time caused by *S. apetala* pericarp and seed extracts compared to the control. The test samples were then compared to diclofenac sodium for central analgesic activity. The crude ME of *S. apetala* pericarp and seed demonstrated notable central analgesic activity at doses of 200 mg/kg bw and 400 mg/kg bw, as shown in Fig 2 and Table S1. The percentage inhibition of elongation at doses of 200 mg/kg and 400 mg/kg by *S. apetala* pericarp and seed MEs were as follows: 3.69 ± 0.15, 5.30 ± 0.22, and 7.40 ± 0.27; 4.50 ± 0.18, 7.29 ± 0.73, and 10.16 ± 0.59; 6.62 ± 0.19, 5.4 ± 0.38, 7.79 ± 0.23; and 4.06 ± 0.23, 6.04 ± 0.42, and 8.18 ± 0.38 at 30 min, 60 min, and 90 min, respectively. The corresponding values for the standard and control at 200 mg/kg and 400 mg/kg doses were 5.55 ± 0.19, 10.12 ± 0.33, 17.84 ± 0.41, and 2.28 ± 0.23, 2.27 ± 0.20, and 2.85 ± 0.27 at 30 min, 60 min, and 90 min, respectively. Significant p-values for MESAP (MEof *S. apetala* pericarp) and MESAS (MEof *S. apetala* seed) of the pericarp and seed at 200 mg/kg bw doses were found to be 0.002, 0.0002, <0.0001; 0.0022, 0.0014, <0.0001 for 30 min, 60 min, and 90 min, respectively, whereas at 400 mg/kg bw, they were 0.0005, 0.0016, 0.003; 0.0001, 0.0002, 0.0004, respectively. The p-values for the standard (diclofenac sodium) were 0.003, <0.0001, and 0.0001 at 30 min, 60 min, and 90 min, respectively, indicating significant differences. As a result, further investigations can be carried out to develop *S. apetala* fruit extract as a central analgesic medication. This fruit extract has the potential to interfere with the binding of prostanoids to receptors or the metabolic pathway leading to prostaglandin formation. Although it is comparable to the standard medication at a dose of 200 mg/kg bw, the maximum effect was observed at a dose of 400 mg/kg bw.

The ME extract may also inhibit neural mediator synthesis [20]. The analgesic efficacy of *S. apetala* seed extracts was evaluated using common solvents such as n-hexane>chloroform>ethyl acetate>methanol> water by their polarity. All extracts significantly decreased acetic acid-induced writhing in mice at 500 mg/kg body weight, outperforming diclofenac sodium (25 mg/kg). However, the n-hexane extract exhibited lower efficacy compared to the positive control. In the hot plate assay, all fractions reduced reaction time, with the diethyl ether fraction maintaining significant analgesic effects for 180 minutes. Additionally, an ethanolic fruit extract study revealed a dose-dependent reduction in writhing, with 46.54% inhibition at 250 mg/kg and 69.62% at 500 mg/kg body weight [21].

Each mouse's writhing count was recorded, and the average count for each group was calculated and is listed in Table 1. The percent inhibition of writhing induced by the standard control, *S. apetala* pericarp, and methanolic seed extracts was found to be 4.5 ± 1.29 and 21 ± 1.82; 11.5 ± 1.91 and 8.5 ± 1.73; and 11.5 ± 1.29 and 9 ± 1.82 at 200 mg/kg and 400 mg/kg, respectively. The test samples were then compared to diclofenac sodium for peripheral analgesic activity. The statistical results for ME and the standard were 0.0007, 0.0018, 0.0028, <0.0001, and <0.0001, showing that ME had significant peripheral analgesic effects at 200 mg/kg and 400 mg/kg doses. This suggests potential for further research to discover

**Table 1. Evaluation of peripheral analgesic activity from crude extract of *S. apetala* pericarp and seed.**

| Group | Dose (mg/kg b.w.) | No. of writhing *MESP* | No. of writhing *MESS* |
|---|---|---|---|
| CTL | 0 | 21 ± 1.82 | 21 ± 1.82 |
| STD | 2 | 4.5 ± 1.29 | 4.5 ± 1.29 |
| MESF | 200 | 11.5 ± 1.91 | 11.5 ± 1.29 |
| MESF | 400 | 8.5 ± 1.73 | 9 ± 1.82 |

new compounds. Analgesic properties of seed extracts were tested with n-hexane, chloroform, and methanol. In the Sundarbans, Ceriops decandra showed the highest analgesic activity, reducing acetic acid-induced writhing by 45% [1,22].

### 3.2 Antidiarrheal activity of the ME of *S. apetala* fruit

The castor oil-induced diarrheal model is a valuable method for studying diarrhea, as autacoids and prostaglandins are known to play a significant role in its development [23,24]. The percentage reduction rates of the ME pericarp and seed at doses of 200 and 400 mg/kg body weight are shown in Fig 3 and Table S2. At 200 mg/kg, the reduction rates were 77.78%, 90.0%, 78.13%, and 67.44%, and at 400 mg/kg, they were 66.67%, 80.0%, 78.13%, and 72.09% at 1 hour, 2 hours, 3 hours, and 4 hours, respectively. Notably, the pericarp of *S. apetala* at a 200 mg/kg dose at 2 hours showed activity almost identical to the standard, while the ME of the seed at a 400 mg/kg dose at 1 hour exhibited the most potent activity. The results demonstrate significant antidiarrheal activity. At 200 and 400 mg/kg doses, standard and control rates were 100 vs. 2.25; 95 vs. 5.0; 97.87 vs. 8.0; and 81.39 vs. 10.45 at 1, 2, 3, and 4 hours, respectively. Castor oil releases ricinoleic acid, which causes inflammation and irritation of the intestinal mucosa, stimulating prostaglandin release and increasing motility and secretion [25].

*S. apetala* seeds and fruits have demonstrated significant efficacy as antidiarrheal agents. In a study by [1], young Swiss albino mice were given seed extracts from different solvent fractions after castor oil-induced diarrhea, showing significant excretion inhibition. Methanol, chloroform, n-hexane, and ethyl acetate fractions inhibited excretion by 68%, 65%,

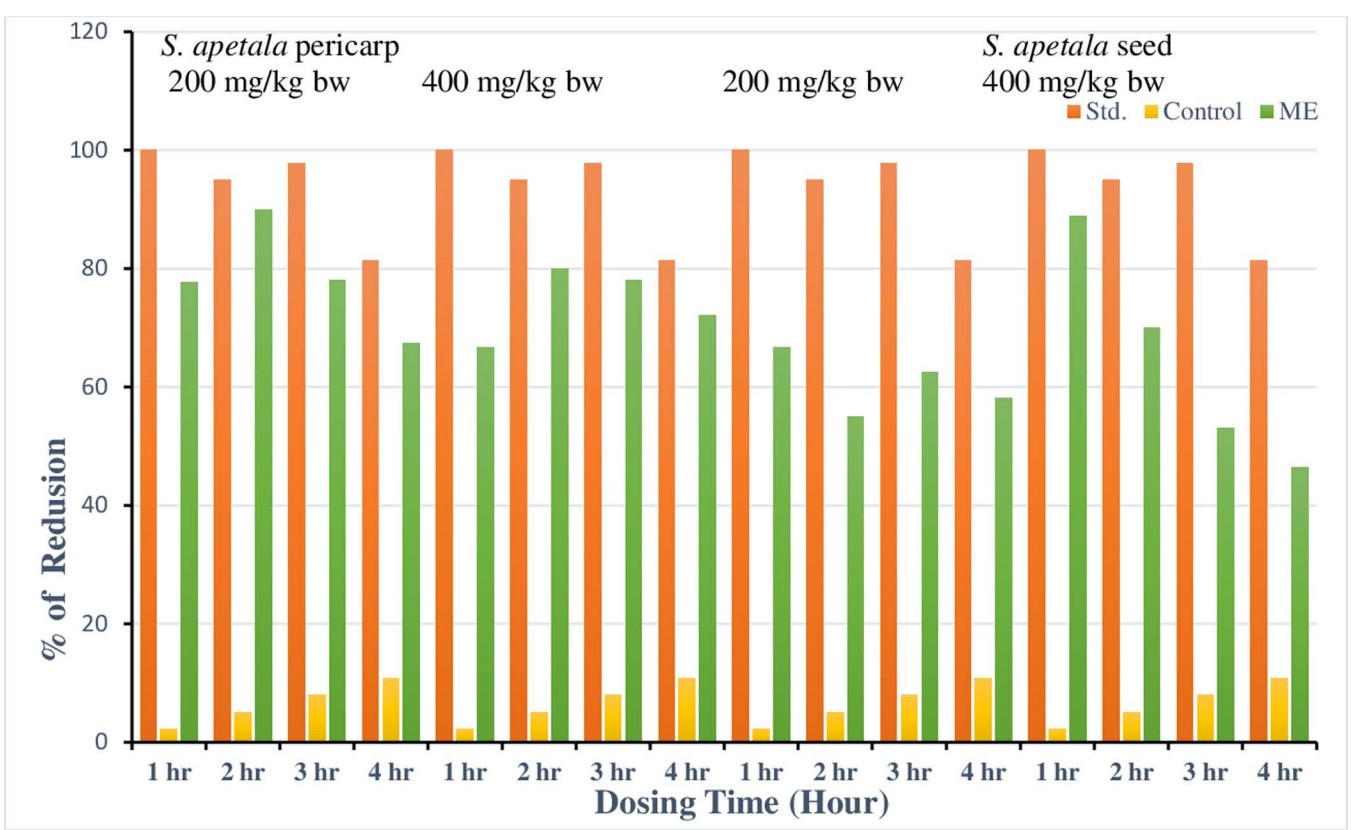

**Fig 3. Antidiarrheal activity of *S. apetala* pericarp and seed extracts evaluated using the castor oil-induced diarrhea model (ricinoleic acid mechanism).**

51%, 50%, and 39%, respectively, at 500 mg/kg. The MEF reduced defecation by 74.19%, 82.26%, 87.90%, and 94.35% at doses of 62.5, 125, 250, and 500 mg/kg, demonstrating strong antidiarrheal effects [1,4].

### 3.3 Antidepression Status of the ME of *S. apetala* fruit

Statistical evaluation demonstrated that MEsderived from *S. apetala* pericarp and seed induced mild central nervous system (CNS) depressive activity at 200 and 400 mg/kg body weight, as illustrated in Fig 4 and Table S3. The standard group exhibited an average sleep onset time (ATOS) of 39.75 minutes and total sleep duration (ATST) of 79.75 minutes, closely mirrored by the control group. For the pericarp extract, ATOS and ATST were 39.25 minutes and 68.5 minutes, respectively, at both dosages. Similarly, the seed extract showed the same ATOS and ATST values. The P values for the pericarp and seed extracts, as well as for the standard, were 0.04, 0.11, 0.07, 0.19, and 0.4390, respectively. Since these values are less than 0.5, the effects were not considered statistically significant. Further studies are needed to explore the potential CNS depressant effects. Additionally, the methanol extract of *S. apetala* bark has shown an average inhibitory effect

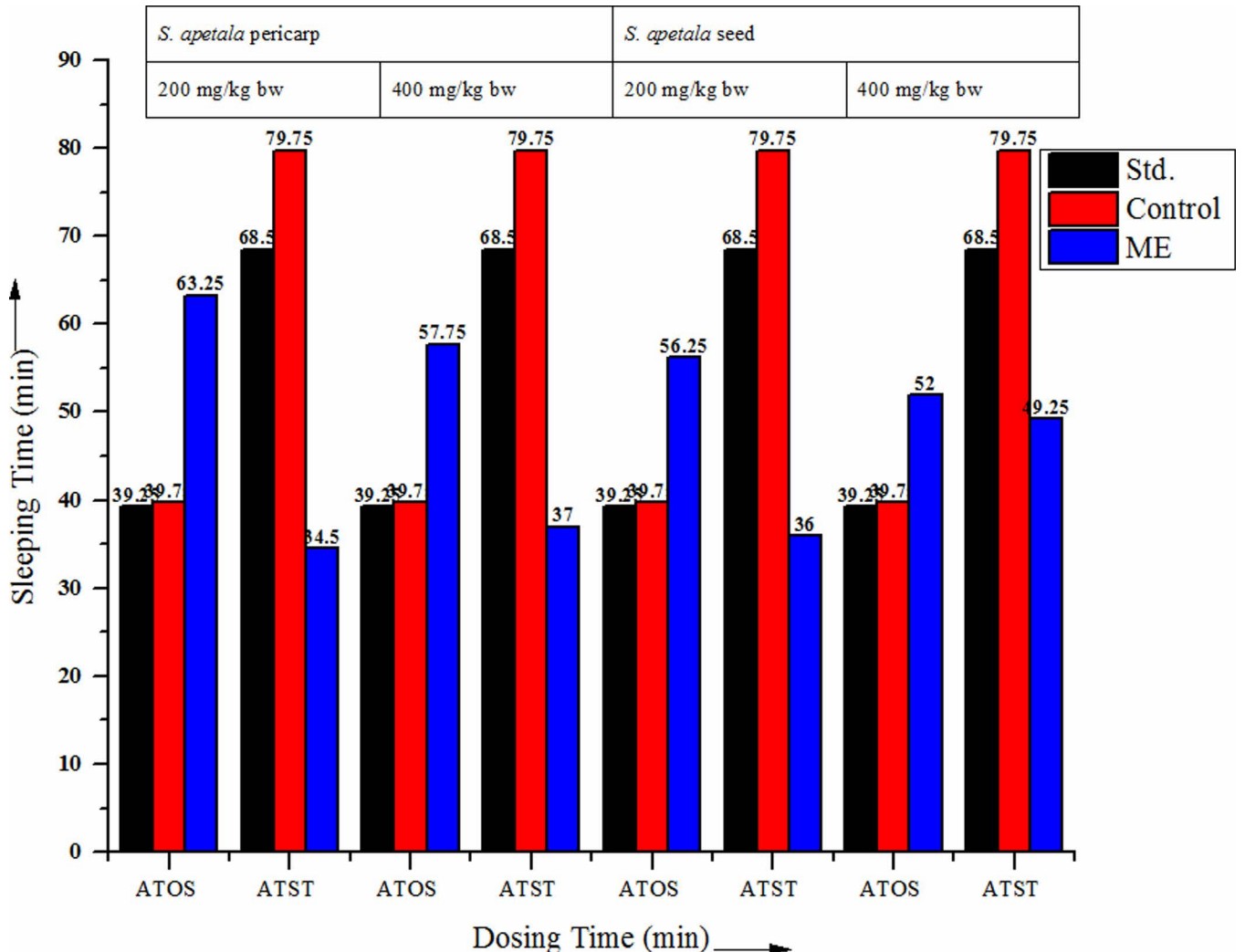

**Fig 4. Antidepressant activity of *S. apetala* pericarp and seed extracts by the thiopental sodium-induced sleeping time method .**

on membrane stability, suggesting its potential anti-inflammatory properties [24,26]. The methanol extract of *S. apetala* bark exhibits anti-inflammatory properties by modulating vascular dilation, immune cell migration, and pro-inflammatory mediator release [27].

### 3.4 Hypoglycemic efficacy of the ME of *S. apetala* fruit

The ME of *S. apetala* pericarp and seed demonstrated notable hypoglycemic activity at doses of 200 and 400 mg/kg body weight, supported by statistically significant findings (Fig 5, Table S4). In a streptozotocin-induced type 2 diabetic rat model, plasma glucose levels in the pericarp-treated group declined from 11.43 mmol/L and 16.43 mmol/L at 60 minutes to 7.35 mmol/L and 10.28 mmol/L at 180 minutes, as depicted in Fig 6. Similarly, in the seed extract-treated group, glucose levels dropped from 10.7 mmol/L and 11.5 mmol/L at 60 minutes to 7.25 mmol/L and 7.68 mmol/L at 180 minutes. Further research is essential to isolate and characterize the specific hypoglycemic compounds responsible for the observed effects. The 99% methanolic extracts (MEs) of *S. apetala* seeds and pericarp demonstrated significant antidiabetic activity, as indicated by reduced serum glucose levels in streptozotocin (STZ)-induced diabetic rats after 135 minutes of administration. The seed extract decreased glucose levels from $13.75 \pm 2.21$ mmol/L to $10.3 \pm 1.75$ mmol/L within 30 minutes, and the pericarp extract reduced levels from $14.36 \pm 2.16$ mmol/L to $11.32 \pm 1.74$ mmol/L. The leaf and bark extracts demonstrated more potent antidiabetic activity, with IC50 values of $0.286 \pm 0.022$ mg/mL and $0.432 \pm 0.01$ mg/mL, respectively, highlighting their significant therapeutic potential [28]. *S. cumini, S. chirata,* and *F. racemosa* bark are rich in antioxidants, offering diverse health benefits and potential for diabetes treatment [29].

### 3.5 Antimicrobial Status of the ME from *S. apetala* fruit

The comparative analysis of antibacterial efficacy between the ME and standard amoxicillin is detailed in Table 2 and Fig S1. At 400 µg/disc, the ME of *S. apetala* seed and pericarp demonstrated inhibition zones of 9–15 mm and 10–13 mm, respectively, indicating superior antibacterial activity in the seed extract. Additionally, both seed and pericarp extracts

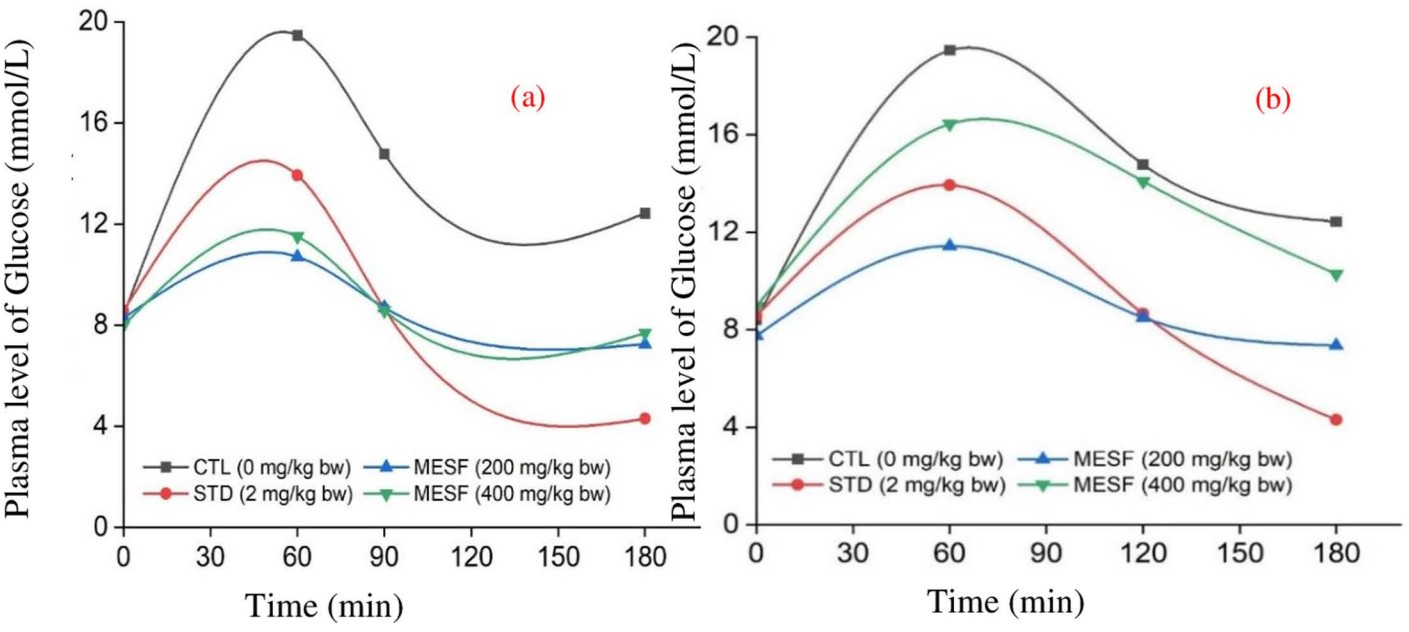

**Fig 5. Plasma level of glucose (mmol/L) of mice at different times for the (a) pericarp and (b) seed of *S. apetala*.**

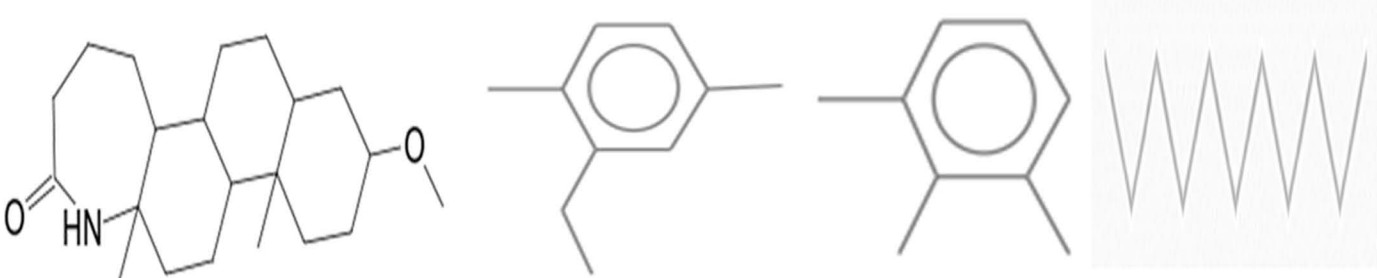

**N-Ethyl-2-methylbenzenesulfonamide    N-Acetyl-appa-aminooxybutyric  acid, methyl    3,6-Pyridazinedione,1,2-dihydro-1-(4-nitrophenyl)**

**2H-Phenanthro[2,1-b]azepin-2-one,  1,3,4,5,5a    Benzene, 2-ethyl-1, 4-diethyl    Benzene, 1,2,3 trimethyl    Undecane**

**Fig 6.  Chemical structures of compounds in seed and pericarp of ME.**

displayed significant antifungal properties, underscoring their potential as antimicrobial agents. These findings highlight the differential bioactivity of *S. apetala* components, with seeds exhibiting stronger inhibitory effects compared to pericarp.

Maximum and minimum zones of inhibitory activity against *Aspergillus niger* and *Candida albicans* respectively (15 mm for *Aspergillus niger* and 10 mm for both seed and pericarp). In gram-negative bacteria, ME of pericarp inhibited *Pseudomonas aeruginosa* and *Salmonella Paratyphi*, whereas *Salmonella Typhi* was inhibited by ME of both seed and pericarp up to 10 mm. For gram-positive bacteria, Sarcinalutea was inhibited by the ME of the pericarp, while Staphylococcus aureus was inhibited by the ME of both the seed and pericarp, as seen up to 10 mm. ME of seeds further showed inhibition of gram positive and gram negative bacteria as well [1] against *Salmonella Typhi, Salmonella Paratyphi A, E. coli, Shigella dysenteriae, S. aureus, Proteus species, Pseudomonas species* and *Shigella flexneri*. The leaf extract of *S. apetala* showed significant antifungal activity against both *Candida albicans* and *Aspergillus flavus* in different solvents, where acetone was the most effective [30,31]. The methanolic seed extract also displayed antibacterial activity against both gram-positive and gram-negative bacteria confirming its traditional use by coastal peoples for treating diarrhea [4]. Bark and leaves have significant antibacterial activity (acetone extracts) [28]. Also, different leaf extracts have indicated antifungal properties, with acetone extract most active against *Aspergillus flavus* and *Candida albicans* [18].

### 3.6 Bioactive constituents of *S. apetala* fruit of n-hexane, ethyl acetate, and chloroform fractions

Seven major compounds were detected in the n-hexane, ethyl acetate and chloroform fractions of *S. apetala* fruit pericarp and seed by GC-MS analysis. These compounds include N-Ethyl-2-methylbenzenesulfonamide, 3,6-Pyridazinedione, 1,2-dihydro-1-(4-nitrophenyl), N-Acetyl-alpha-aminooxybutyric acid (methyl), 2H-Phenanthro [2,1-b] azepin-2-one (1,3,4,5,5a), Benzene (2-ethyl-1,4-diethyl), Benzene (1,2,3-trimethyl), and Undecane. GC-MS analysis identified seven distinct compounds

**Table 2. Antimicrobial activity of the extracts (MEs) from the pericarp and seed of *S. apetala*.**

| Microbial Name | *Std.* Amoxicillin (30 µg/disc) | Methanol extract (400 µg/disc) | |
| --- | --- | --- | --- |
| | | seed | Pericarp |
| Gram Positive bacteria | | | |
| *Bacillus megaterium,* | *38* | – | – |
| *Bacillus subtilis* | *40* | – | – |
| *Bacillus sereus* | *40* | – | – |
| *Staphylococcus aureus,* | *40* | *9* | *10* |
| *Sarcinalutea* | *45* | – | *10* |
| Gram Negative bacteria | | | |
| *Escherichia coli,* | *40* | – | – |
| *Vibrio mimicus,* | *37* | – | – |
| *Shigelladysenteriae,* | *45* | – | – |
| *Pseudomonas aeruginosa,* | *45* | – | *10* |
| *Shigella boydii,* | *35* | – | – |
| *Salmonella Paratyphi,* | *40* | – | *10* |
| *Salmonella Typhi,* | *48* | *10* | *10* |
| Fungi | | | |
| *Sacharomycescerevacae,* | *38* | – | *12* |
| *Candida albicans,* | *45* | *10* | *10* |
| *Aspergillus niger* | *40* | *15* | *13* |

in the n-hexane, ethyl acetate, and chloroform pericarp extracts of *Sonneratia apetala* (see Fig 6 and Table 3). The structures of these compounds were visualised using ChemDraw software, as illustrated in Fig 6 (refer to Table 3). The chromatogram for the n-hexane, ethyl acetate, and chloroform seed fractions revealed six prominent peaks, corresponding to the compounds N-Ethyl-2-methylbenzenesulfonamide, 3,6-Pyridazinedione, 1,2-dihydro-1-(4-nitrophenyl), N-Acetyl-alpha-aminooxybutyric acid (methyl), 2H-Phenanthro[2,1-b] azepin-2-one (1,3,4,5,5a), and Undecane, with retention times of 21.38, 16.29, 22.63, 17.48, and 12.3 minutes, respectively. In comparison, the pericarp extract chromatogram indicated three major peaks, while both ethyl acetate and chloroform fractions in the seed and pericarp extracts exhibited two distinct peaks. These included Benzene (2-ethyl-1,4-diethyl) and Benzene (1,2,3-trimethyl), with retention times of 11.56 and 8.6 minutes, respectively. Table 3 presents the structural details of the predominant compounds identified in the n-hexane extract of *S. apetala* fruit.

N-Ethyl-2-methylbenzenesulfonamide is recognised for its bioactivity, particularly as a potential antimicrobial and antifungal agent [32,33]. This compound has shown efficacy in disrupting bacterial cell wall synthesis and fungal cell membrane integrity, making it valuable in studies targeting resistant microbial strains. Additionally, its sulfonamide structure

**Table 3. Tentative compounds in *S. apetala* ME were identified via GC-MS.**

| SI No | Compounds Name | Fraction | M. weight | M. formula | P. Area | m/z | RT |
| --- | --- | --- | --- | --- | --- | --- | --- |
| 01 | •N-Ethyl-2-methylbenzenesulfonamide | nHSF | 199.270 | $C_9H_{13}NO_2S$ | 3128 | 813 | 21.38 |
| 02 | 3,6-Pyridazinedione, 1,2-dihydro-1-(4-nitrophenyl) | nHSF | 233.18 | $C_{10}H_7N_3O_4$ | 5567 | 387 | 16.29 |
| 03 | N-Acetyl-appa-aminooxybutyric acid, methyl | nHSF | 386.39 | $C_{16}H_{17}F_3N_4O_2S$ | 2679 | 656 | 22.63 |
| 04 | 2H-Phenanthro[2,1-b]azepin-2-one, 1,3,4,5,5a | nHSF | 333.5 | $C_{21}H_{35}NO_2$ | 3180 | 164 | 17.48 |
| 05 | Benzene, 2-ethyl-1, 4-diethyl | CSF, CFF | 134.22 | $C_{10}H_{14}$ | 3165 | 119 | 11.56 |
| 06 | Benzene, 1,2,3 trimethyl | EASF EAFF | 120.19 | $C_9H_{12}$ | 4733 | 105 | 8.6 |
| 07 | Undecane | nHFF | 156.31 | $C_{11}H_{24}$ | 2946 | 57 | 12.3 |

allows it to inhibit certain enzymes involved in folate synthesis, a pathway critical for cell division and growth in microbes, thereby exhibiting therapeutic potential in antimicrobial drug development.

3,6-Pyridazinedione, 1,2-dihydro-1-(4-nitrophenyl) exhibits promising bioactivity, particularly as an anti-inflammatory and antioxidant agent [34–36]. Pyridazinedione core structure is likely responsible for the observed capability of inhibiting reactive oxygen species (ROS) and modulating inflammatory pathways, which has attracted the attention of researchers studying diseases associated with oxidative stress and inflammation. Pyridazinedione derivatives have proven to inhibit different key enzymes and pathways involved in inflammatory processes such as COX-2 and NF-kB, which are anti-inflammatory drug targets. These properties indicate the potential of this compound to overcome chronic inflammatory diseases and oxidative damage.

Due to the aminooxy functional group, N-Acetyl-alpha-aminooxybutyric acid (methyl) has been shown to have bioactivity as an enzyme inhibitor. This allows for parasite growth inhibition by targeting PLP-dependent enzymes involved in amino acid metabolism. Inhibition such as this can be useful in biochemistry, where the blocking of certain metabolic pathways allows scientists to gain better information about the metabolism itself by inhibiting individual enzyme functions. Furthermore, it has been shown that derivatives of aminooxy, such as N-Acetyl-alpha-aminooxybutyric acid(methyl), exhibit antimicrobial and anti-cancer properties [37]. They can also be used to change biological reactions in cells, so they might have therapeutic applications, but much remains to be explored to get insight into their effectiveness and mechanisms of action, with studies being conducted up to our data cutoff in October 2023.

A substantial bioactivity as potential anticancer and antimicrobial agents have been reported for 2H-Phenanthro [2,1-b] azepin-2-one (1,3,4,5,5a) [38]. Its novel azepine scaffold is capable of binding to DNA and impairing the replication process, and it has shown cytotoxic activity against some cancer cell lines. It has also been shown to inhibit important bacterial enzymes that are considered potential targets for antimicrobials. However, studies suggest that this compound has the potential to induce apoptosis in cancerous cells due to its mitochondrial activity, making it a candidate for use in oncology and a potential treatment for infectious disease.

The alkyl-substituted benzene delivering limited direct bioactivity, benzene, 2-ethyl-1,4-diethyl is a common structural analogue that is used in hydrocarbon bioactivity and environmental health studies [39]. It is an established practice to evaluate toxicological profiles of aliphatic-substituted benzene derivatives, as a benzene ring substituted with alkyl moieties can cause complications in the metabolic pathway results, compensated for by a production of potentially hazardous metabolite. The noteworthy actions of these compounds in hepatic enzyme induction, oxidative stress, and cellular toxicity have furthered research on environmental pollutants and health effects. Marinas et al. recently identified hemimellitene (or benzene (1,2,3-trimethyl)), known for its bioactivity, including hepatotoxicity and neurotoxicity [40], owing to its activation to reactive intermediates that generate oxidative stress and lead to DNA damage. As a methylated derivative of benzene, this compound has toxicological importance for initiating changes in liver enzyme activity as well as inducing the formation of free radicals and cell damage..it is also being investigated regarding its impact on respiratory health and as a potential Environmental Pollutant of Particulate low significance in occupational and environmental health.

Undecane a straight-chain alkane is mainly studied as a solvent and in environmental-related studies regarding its hydrophobicity, and non-reactivity [28,41]. Although undecane has poor bioactivity, it can readily function as a volatile organic compound (VOC) and promote atmospheric pollution with long-term exposure effects on the respiratory system. Undecane is also of interest in studies of insect behaviour and is known to act as a pheromone or signalling compound in some species of insects, influencing their aggregation and mating behavior.

### 3.6.1 Potential mechanisms of action of the bioactive compounds identified in *S. apetala* extracts.

These bioactive compounds include N-Ethyl-2-methylbenzenesulfonamide, 3,6-Pyridazinedione (1,2-dihydro-1-(4-nitrophenyl)), N-Acetyl-alpha-aminooxybutyric acid (methyl), 2H-Phenanthro[2,1-b] azepin-2-one (1,3,4,5,5a), Benzene (2-ethyl-1,4-diethyl), Benzene (1,2,3-trimethyl), and Undecane, which are discussed below, with potential mechanisms of action based on their chemical properties.

Enzymatic pathways susceptible to N-Ethyl-2-methylbenzenesulfonamide include sulfation pathways; this gas may also inhibit pathways of nucleic acid synthesis or pathways toward oxidative stress because of its nitrogen-containing nitrophenyl group (3,6-Pyridazinedione). N-Ethyl-2-methylbenzenesulfonamide (mimics PABA) and reacts with dihydropteroate synthase (DHPS) [42]:

$$\text{Inhibition of Folic Acid Biosynthesis PABA + DHPS} \rightarrow$$

Nucleophilic aromatic substitution (electron-deficient nitrophenyl) and Nu = nucleophile, attacks electron-poor ring due to nitro group [43].

$$C_6H_4(NO_2) + Nu{-} \rightarrow C_6H_4(NO_2)\,Nu$$

Act as an Electron Acceptors and Disruption of the Oxidative Stress [44]
Nitro Group Germanium Oxide Group → Radical Intermediates → Oxidative Stress
Pyridoxal phosphate dependent enzymes related to amino acid metabolism could be inhibited by N-Acetyl-alpha-aminooxybutyric acid. It is reacted with the aminooxy group ($-ONH_2$) and forms a Schiff base with pyridoxal phosphate (PLP) (Vitamin B6 cofactor) [45]. This inhibits PLP-dependent enzymes like transaminases.

$$R{-}CH{=}O + H2N{-}O{-}R \rightarrow R{-}C{=}N{-}O{-}R$$

The polycyclic structure of 2H-Phenanthro [2,1-b] azepin-2-one allows it to work at these different pathways like a DNA intercalated or topoisomerase inhibitor. DNA intercalation inhibits replication and stabilizes the DNA-cleavage complex, blocking re-ligation [46].

$$(7)\ \text{Polycyclic Ring} \rightarrow \text{Decrease in DNA Helix Stability}$$

Its aromatic hydrocarbons, for example, 2-ethyl-1,4-diethyl benzene, benzene, 1,2,3-trimethyl may induce cell membranes disruption or reactive oxygen species generation. Reactive intermediates deriving from oxidative metabolism and Cyto-chrome P450 enzymes oxidize the alkylated benzenes [47]:

$$R{-}C6H4{-}R' + O2\ R{-}C6H3{-}OH + ROS.$$

This creates reactive oxygen species (ROS) that lead to cellular damage.

As saturated alkane Undecane may also affect membrane integrity in high concentrations. Undecane, a long-chain hydrocarbon, incorporates into lipid bilayers and disrupts membrane fluidity [48,49]:

$$\text{Membrane Permeability Alteration} \rightarrow \text{Undecane + Lipid Bilayer}$$

These mechanisms can have diverse applications, such as inhibition of enzymes [40], interaction with DNA [41,42], disruption of membranes [43] and induction of oxidative stress [44,45].

## 5 Conclusion

Based on the findings of this study, *S. apetala* fruit demonstrates significant bioactivity with applications in traditional medicine and potential as a source for novel therapeutic agents. The MEs from both seed and pericarp displayed robust analgesic, antidiarrheal, hypoglycaemic, and antifungal properties, aligning with the fruit's established uses in ethnomedicine. GC-MS analysis revealed the presence of these bioactive compounds in the extract, including N-Ethyl-2-methylbenzenesulfonamide and 3,6-Pyridazinedione, which exhibit antimicrobial and anticancer properties. In vivo studies suggest antidepressant effects, robust central and peripheral analgesia, and antidiarrheal activity. The extracts also showed some antibacterial activity against gram-positive and gram-negative bacteria. Moreover, this antifungal activity test showed that ME fraction

of seed and pericarp of *S. apetala* has high antifungal activity. Some gram-positive (e.g., Sarcinalutea and Staphylococcus aureus) and gramnegative (e.g., *Salmonella Paratyphi, Pseudomonas aeruginosa and Salmonella Typhi*) bacteria were found to have some inhibition activities against the ME of the seed and pericarp. The appearance of such activities of other herbs may stimulate to investigate plant materials for their possibly undiscovered efficacy to substantiate their use as traditional medicines and would ultimately lead to the identification of novel lead chemicals owing to their bioactivity.

These findings suggested that *S. apetala* provides a potential natural source of bioactive compounds with various pharmacological effects. Additional studies may isolate specific compounds and investigate their diverse mechanisms and synergetic therapeutic potential. The results suggest *S. apetala* as a potential source of direct-acting plant-based approaches for different kinds of medicinal applications, that could be of great value in medicinal science and serve as a natural substitute for allopathic medicines.

## Supporting information

**Fig S1. Antimicrobial activity of methanolic extracts of the pericarp and seed of *S. apetala.***
(DOCX)

**Table S1. Evaluation of the central analgesic activity of a crude extract of *Sonneratia apetala* pericarp and seed.**
(DOCX)

**Table S2. Effect of methanolic extract of *S. apetala* pericarp and seed on castor oil induced diarrhea in mice.**
(DOCX)

**Table S3. Anti-depressant activity of *S. apetala* by Thiopental sodium induced sleeping time method for pericarp and seed.**
(DOCX)

**Table S4. Plasma level of glucose (mmol/L) of mice at different times for the pericarp and seed of S. apetala.**
(DOCX)

## Acknowledgments

The authors express deep gratitude to INARS, BCSIR, Dhaka, Bangladesh, and the Ministry of Science and Technology, Bangladesh, for their analytical, technical, and logistical assistance. They also thank King Khalid University's Deanship of Scientific Research for funding via grant RGP 2/25/46.

## Author contributions

**Conceptualization:** Md. Ripaj Uddin.

**Data curation:** Md. Ripaj Uddin.

**Formal analysis:** Fatema Akhter, Md. Ripaj Uddin, Muhammad Abdullah Al Mansur, Mohammad Saydur Rahman, Abubakr M. idris, AHM Shofiul Islam Molla Jamal.

**Investigation:** Fatema Akhter, Md. Ripaj Uddin, Muhammad Abdullah Al Mansur, Md. Ahedul Akbor, Nahida Akter, Abubakr M. idris, AHM Shofiul Islam Molla Jamal.

**Methodology:** Fatema Akhter, Md. Ripaj Uddin, Md. Ahedul Akbor, Nahida Akter, Abubakr M. idris.

**Project administration:** Md. Ripaj Uddin.

**Software:** Md. Ripaj Uddin, Md. Golam Mostafa.

**Supervision:** Md. Ripaj Uddin, Mohammad Saydur Rahman.

**Validation:** Md. Ripaj Uddin.

**Visualization:** Md. Ripaj Uddin.

**Writing – original draft:** Fatema Akhter, Md. Ripaj Uddin.

**Writing – review & editing:** Md. Ripaj Uddin, Muhammad Abdullah Al Mansur, Nahida Akter, Abubakr M. idris, Sarker Kamruzzaman.

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
