## [Decision Letter · Decision Letter 0]

21 Jan 2025

PONE-D-24-48754

Antidiarrheal, Antidepressant, Hypoglycemic, Analgesic, and Antimicrobial Activities of Methanolic Extract from Sonneratia apetala Fruit, with Identification of Bioactive Compounds in n-Hexane, Chloroform, and Ethyl Acetate Fractions

PLOS ONE

Dear Dr. Uddin,

Thank you for submitting your manuscript to PLOS ONE. After careful consideration, we have decided that your manuscript does not meet our criteria for publication and must therefore be rejected.

Specifically:

I agree the review that your results� at currrent stage, could not be supportted by the data, more data are necessary. 

I am sorry that we cannot be more positive on this occasion, but hope that you appreciate the reasons for this decision.

Kind regards,

Xiaoshan Zhu, Ph.D.

Academic Editor

PLOS ONE

Reviewers' comments:

Reviewer's Responses to Questions

**Comments to the Author**

1. Is the manuscript technically sound, and do the data support the conclusions?

Reviewer #1: Yes

Reviewer #2: No

2. Has the statistical analysis been performed appropriately and rigorously? 

Reviewer #1: Yes

Reviewer #2: N/A

3. Have the authors made all data underlying the findings in their manuscript fully available?

Reviewer #1: No

Reviewer #2: No

4. Is the manuscript presented in an intelligible fashion and written in standard English?

Reviewer #1: Yes

Reviewer #2: No

5. Review Comments to the Author

Reviewer #1: In this manuscript, the authors have investigated the potential medicinal properties of Sonneratia apetala fruit, a mangrove species native to Bangladesh. The study focuses on evaluating the analgesic, antidiarrheal, antidepressant, hypoglycemic, and antimicrobial activities of methanolic extracts from both the pericarp and seed of the fruit. The authors also aim to identify the bioactive compounds present in these extracts.

This research addresses a relevant topic, as the search for novel therapeutic agents from natural sources is crucial, especially with the rise of antimicrobial resistance.

This study employs a comprehensive approach by investigating a wide range of bioactivities, including analgesic, antidiarrheal, antidepressant, hypoglycemic, and antimicrobial effects. This provides a thorough assessment of the fruit's medicinal potential.

The use of standard agents like diclofenac sodium, loperamide, diazepam, and glibenclamide as controls strengthens the validity of the bioactivity assays.

The study also follows ethical guidelines by obtaining approval from the Institutional Animal Ethics Committee.

The use of GC-MS analysis enables the identification of specific bioactive compounds, providing insights into the chemical basis for the observed bioactivities.

However, the manuscript lacks detailed information on the extraction process, such as the solvent-to-sample ratio, extraction time, and the yield of the extracts. This information is crucial for reproducibility and for assessing the efficiency of the extraction method.

The antidepressant activity assessment relies solely on the thiopental sodium-induced sleep time test, which is not a comprehensive measure of antidepressant effects. Including other behavioral tests, such as the forced swim test or tail suspension test, would provide a more robust evaluation of antidepressant potential.

While the antimicrobial activity is assessed against several bacterial and fungal strains, the manuscript doesn't provide the minimum inhibitory concentration (MIC) values, which are essential for determining the potency of the antimicrobial effects

The discussion section could be strengthened by providing more in-depth analysis and comparison of the findings with previous studies on S. apetala and other medicinal plants.

The manuscript would benefit from a more detailed discussion of the potential mechanisms of action of the bioactive compounds identified in the GC-MS analysis. Relating the identified compounds to their potential roles in the observed bioactivities would enhance the scientific significance of the findings

Reviewer #2: The present manuscript contains 30% similarity and 29% AI generated material.

I recommend authors to clear similarity and AI text and re-submit the article.

This manuscript cannot be proceed and published in present form.

6. PLOS authors have the option to publish the peer review history of their article (what does this mean? ). If published, this will include your full peer review and any attached files.

**Do you want your identity to be public for this peer review?** For information about this choice, including consent withdrawal, please see our Privacy Policy .

Reviewer #1: **Yes: ** Shafi Ullah Khan

Reviewer #2: No

- - - - -

---

## [Author Response · Author response to Decision Letter 1]

12 Feb 2025

Reviewer #1:

Q1: In this manuscript, the authors have investigated the potential medicinal properties of Sonneratia apetala fruit, a mangrove species native to Bangladesh. The study focuses on evaluating the analgesic, antidiarrheal, antidepressant, hypoglycemic, and antimicrobial activities of methanolic extracts from both the pericarp and seed of the fruit. The authors also aim to identify the bioactive compounds present in these extracts. This research addresses a relevant topic, as the search for novel therapeutic agents from natural sources is crucial, especially with the rise of antimicrobial resistance. This study employs a comprehensive approach by investigating a wide range of bioactivities, including analgesic, antidiarrheal, antidepressant, hypoglycemic, and antimicrobial effects. This provides a thorough assessment of the fruit's medicinal potential. The use of standard agents like diclofenac sodium, loperamide, diazepam, and glibenclamide as controls strengthens the validity of the bioactivity assays. The study also follows ethical guidelines by obtaining approval from the Institutional Animal Ethics Committee.

Response: Thank you for your nice complement.

Q2: The use of GC-MS analysis enables the identification of specific bioactive compounds, providing insights into the chemical basis for the observed bioactivities. However, the manuscript lacks detailed information on the extraction process, such as the solvent-to-sample ratio, extraction time, and the yield of the extracts. This information is crucial for reproducibility and for assessing the efficiency of the extraction method.

Response: Thank you for your nice observation. In the revised manuscript, we have provided the detailed information on the extraction process (Solvent extraction method), such as the solvent-to-sample ratio (10:1), extraction time (72 hr), and the yield of the extracts (10 gm) at sub section 2.3 in LN 118-140.

Q3: The antidepressant activity assessment relies solely on the thiopental sodium-induced sleep time test, which is not a comprehensive measure of antidepressant effects. Including other behavioral tests, such as the forced swim test or tail suspension test, would provide a more robust evaluation of antidepressant potential.

Response: Thank you for your nice query. We perform Central analgesic activity and Peripheral analgesic activity at subsection 2.6 and 2.7 that is why forced swim test or tail suspension test was not performance. In other work, we will try to cover this test more robust evaluation of antidepressant potential.

Q4: While the antimicrobial activity is assessed against several bacterial and fungal strains, the manuscript doesn't provide the minimum inhibitory concentration (MIC) values, which are essential for determining the potency of the antimicrobial effects.

Response: Thank you for your nice suggestion. In the revised manuscript, we have provided the minimum inhibitory concentration (MIC) values of of ciprofloxacin is 0.015 - 2 µg/mL at subsection 2.11 in LN 211.

Q5: The discussion section could be strengthened by providing more in-depth analysis and comparison of the findings with previous studies on S. apetala and other medicinal plants.

Response: Thank you for your nice suggestion. In the revised manuscript, we have provided more in-depth analysis and comparison of the findings with previous studies on S. apetala and other medicinal plants at section 3 in LN 271-274, 303-306, 316-317, 328-330, 354-356, 371-373, and 395-401.

Q6: The manuscript would benefit from a more detailed discussion of the potential mechanisms of action of the bioactive compounds identified in the GC-MS analysis. Relating the identified compounds to their potential roles in the observed bioactivities would enhance the scientific significance of the findings.

Response: Thank you for your nice suggestion. In the revised manuscript we have added detailed discussion of the potential mechanisms of action of the bioactive compounds identified in the GC-MS analysis at LN 475-509 in sub section 3.5.1.

Reviewer #2:

The present manuscript contains 30% similarity and 29% AI generated material.

I recommend authors to clear similarity and AI text and re-submit the article.

This manuscript cannot be proceed and published in present form.

Response: In the revised manuscript, the similarity index is less than 20% according to iThenticate software and 0% according to Turnitin software. All AI-generated material has been removed, and the AI report shows 0%.

---

## [Editor Report · Decision Letter 1]

4 Mar 2025

Antidiarrheal, Antidepressant, Hypoglycemic, Analgesic, and Antimicrobial Activities of Methanolic Extract from Sonneratia apetala Fruit, with Identification of Bioactive Compounds in n-Hexane, Chloroform, and Ethyl Acetate Fractions

PONE-D-24-48754R1

Dear Dr. Uddin,

We’re pleased to inform you that your manuscript has been judged scientifically suitable for publication and will be formally accepted for publication once it meets all outstanding technical requirements.

Kind regards,

Hope Onohuean, PhD

Academic Editor

PLOS ONE
---

## [Editor Report · Acceptance letter]

PONE-D-24-48754R1

PLOS ONE

Dear Dr. Uddin,

I'm pleased to inform you that your manuscript has been deemed suitable for publication in PLOS ONE. Congratulations! Your manuscript is now being handed over to our production team.

Kind regards,

on behalf of

Dr. Hope Onohuean

Academic Editor

PLOS ONE